# Deep learning model for fully automated breast cancer detection system from thermograms

**Esraa A. Mohamed**[1], **Essam A. Rashed**[1,2], **Tarek Gaber**[3,4]*, **Omar Karam**[5]

1 Department of Mathematics, Faculty of Science, Suez Canal University, Ismailia, Egypt, 2 Department of Electrical and Mechanical Engineering, Nagoya Institute of Technology, Nagoya, Japan, 3 School of Science, Engineering and Environment University of Salford, Manchester, United Kingdom, 4 Faculty of Computers and Informatics, Suez Canal University, Ismailia, Egypt, 5 Faculty of Informatics and Computer Science, British University in Egypt (BUE), Cairo, Egypt

* t.m.a.gaber@salford.ac.uk

## Abstract

Breast cancer is one of the most common diseases among women worldwide. It is considered one of the leading causes of death among women. Therefore, early detection is necessary to save lives. Thermography imaging is an effective diagnostic technique which is used for breast cancer detection with the help of infrared technology. In this paper, we propose a fully automatic breast cancer detection system. First, U-Net network is used to automatically extract and isolate the breast area from the rest of the body which behaves as noise during the breast cancer detection model. Second, we propose a two-class deep learning model, which is trained from scratch for the classification of normal and abnormal breast tissues from thermal images. Also, it is used to extract more characteristics from the dataset that is helpful in training the network and improve the efficiency of the classification process. The proposed system is evaluated using real data (A benchmark, database (DMR-IR)) and achieved accuracy = 99.33%, sensitivity = 100% and specificity = 98.67%. The proposed system is expected to be a helpful tool for physicians in clinical use.

## 1. Introduction

Breast cancer is one of the most commonly diagnosed malignancies in women around the world [1]. In 2018, breast cancer reached approximately 15% of registered cases of cancer-linked death among women [2, 3]. Breast abnormalities can be detected by self-examination, physicians, or imaging techniques. The only way to assure whether there is cancer or not is biopsy [4]. There are several breast imaging techniques (for examples ultrasound, mammography. . . etc), which are currently being used for early detection of breast cancer [5]. The leading and most popular screening modality is the mammography due to the relatively high-accuracy, low-cost, and high detectability [6, 7]. Mammograms can provide an effective imaging tool for high accuracy for breast cancer detection and classification. However, its performance is known to be weak in some cases especially for patients with dense breast tissues [8]. Moreover, it may lead

**Data Availability Statement:** All relevant data are within the paper and its Supporting Information files.

**Funding:** The author(s) received no specific funding for this work.

**Competing interests:** The authors have declared that no competing interests exist.

to sever side-effects related to ionized radiation for young age ladies [9]. Moreover, it is known that observing small size lesion less than 2 *mm* is difficult using mammograms [10]. These limitations lead to a high interest in thermography, which is an emerging technology in breast cancer screening. Thermography is a radiation-free, low-cost, non-inclusive, and non-invasive technique [11]. Therefore, it can be used to detect early-stage breast cancer in young women and individuals with dense breasts.

The main idea of thermography is that all living bodies emit infrared (IR) above absolute zero [1, 8]. A thermal infrared camera converts IR radiation into electrical signals, which are shown as a thermogram, in the breast thermography modality [12]. Therefore, potential abnormalities are emphasized and separated from normal tissue as it has a different temperature scale [13, 14]. Breast thermography has several advantages over mammography, including its ability to work with dense breast tissues, effectiveness across all age groups, and ease of use for male patients [5]. Thermography is known for being safe (non-ionized radiation), quick, and leads to early detection of breast cancer [5]. Fig 1 presents the procedure for breast thermography.

Breast area segmentation is a technique for separating the breast region from other parts of the body in thermal images, is an important step in any breast cancer detection system [15]. As much as possible, the extracted region must include all breast tissues, ducts, lobules and lymph nodes. Breast segmentation process ranges from a totally manual to a fully automatic. Because of the unique properties of each breast, which make them amorphous, and the lack of clear boundaries in this type of images, most scientific researches prefer to extract the breast region by using manual or semi-automatic extraction process.

During the last decades, scientific researches were focused on machine learning methods concerned with the diagnosis of breast cancer using thermography; some researchers concentrate their work on determining the size and location of tumors; but others have been concentrated on characteristics such as acquisition protocols and breast quadrants. Deep learning is one of machine learning methods, which uses multilayer convolutional neural networks (CNN) [16]. Deep learning has the ability to automatically extract features from a training dataset [7]. In recent years, scientists have achieved promising results with CNNs for the diagnosis of breast cancer. In the past, the usage of CNNs for the diagnosis of breast cancer with thermal images was not widely used, maybe because of the efficiency of CNNs in comparison with texture or statistical features, or because of the high of computational load [17]. In recent years, CNNs were considered as one of the leading methods for pattern recognition.

The thermal image contains incorporates superfluous areas as neck, shoulder, chess and other parts of the body which behaves as noise during the training in CNN models. However, thermography images are difficult to process due to low-resolution in image spatial domain, it is necessary to extract the breast area from the thermal images which considered as a critical task as the results of the classification process are highly depended on segmentation results.

As previously mentioned, breast cancer is considered one of the leading causes of death among women. Therefore, early detection is necessary to save lives. Thermography imaging is an effective diagnostic technique which is used for breast cancer detection with the help of infrared technology, but it is dependent on the radiologist's ability to interpret the thermogram. To the best of knowledge, the prior work has some limitations such as: (1) the limitation of the dataset, (2) some researches of the related work did not consider segmentation of the breast area before classification or extract the breast area manually, (3) some segmentation models removed parts of the breast, and (4) some researches evaluate their model by calculating the accuracy metric only. However, if the dataset is unbalanced, model's high accuracy rate does not guarantee its ability to discriminate distinct classes equally [18]. Therefore, a fully automatic breast cancer detection system from thermograms is needed to diagnose the disease.

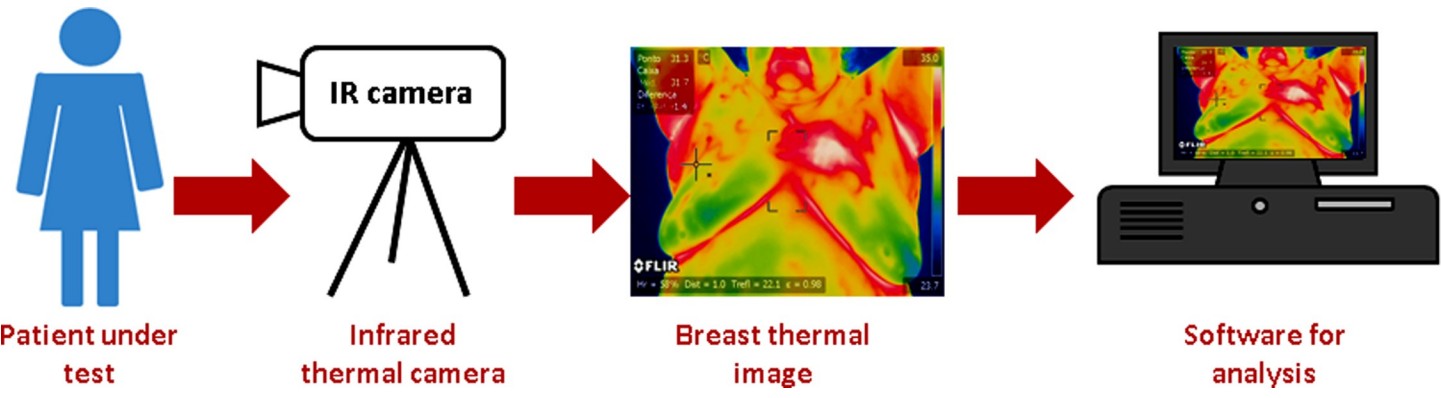

**Fig 1. Breast thermography procedure (thermal image is aquired at room temperature = 22˚C).**

In this study, we propose a fully automatic breast cancer detection system. First, U-Net network is utilized to automatically extract and isolate the breast area from the rest of the body in thermograms. Second, we propose a deep learning model, which is trained for the classification of abnormal breast tissues using thermal images. The proposed method consists of three main phases, resizing, breast area segmentation and deep learning model for classification. In resizing phase, the thermal images are resized to a smaller size to accelerate computation. In breast area segmentation phase, the breast region is extracted automatically by using U-Net network. In deep learning model for classification phase, we proposed a deep learning model based two-class CNN, which is trained from scratch and used for the classification of normal and abnormal breast tissue.

The main contribution of this paper is as following:

1. Extracting and isolating the breast area automatically from other parts of thermal images by using CNN (U-Net).

2. Proposing a deep learning model for the classification of normal and abnormal breast tissues from thermograms

3. Evaluating the performance of the proposed model using accuracy, sensitivity and specificity.

4. Comparing the proposed model with state-of art methods.

The structure of the paper is as follows. Section 2 explains the literature review and Section 3 explains the proposed method. Section 4 contains the experimental results. Finally, the paper is discussed in section 5 and concluded in section 6.

## 2. Literature review

The majority of efforts related to the diagnosis of breast cancer from thermogram use the web available DMR-IR database [19]. In this section, we will present a review of some studies using the DMR-IR database.

Any Computer-Aided Detection (CAD) system for breast cancer detection can be separated into principally three phases: segmentation process, feature extraction and classification. The thermal image contains incorporates superfluous areas as neck, shoulder, chess and other parts of the body, but during the training in CNN models or during the feature's identification process, this data acts as noise. Therefore, several authors have focused their research on decrease as much non-relevant information as possible and extracting region of interest (ROI)

instead of identifying patterns in thermograms. Mahmoudzadeh et al. [20] used extended hidden Markov models (EHMM), BayesNet and Random Forest for the optimization of breast segmentation techniques. But, the proposed method can be used only as a first stage in automatic or semi-automatic system. Also, the of the algorithm need to be improved in case of online application. Ali et al. [1] proposed an automatic segmentation method for ROI extraction from breast thermograms based on the normal and abnormal breasts based on statistical and texture features extracted from ROI. But, the presented method has a limitation of dataset. Also, by this method some lower parts of the breast will be removed. Gaber et al. [8] Proposed an enhanced segmentation method based on both Neutrosophic sets (NS) and optimized Fast Fuzzy c-mean (F-FCM) algorithm. Then, they used different kernel functions of Support Vector Machine (SVM) to detect normal and abnormal breast. They obtained accuracy = 92.06%, recall = 96.55% and precision = 87.50%. But, the proposed segmentation method implemented on a limited number of dataset.

The main process in CAD system is Feature Extraction. This aims to extract certain features from a breast thermogram, analyze and compare these features to obtain significant results. This process will reduce the complexity of classification process. Araujo et al. [21] presented a symbolic data analysis on 50 patients' thermograms and obtained the interval data in the symbolic data analysis and statistical analysis. They proposed three-stage feature extraction method. In the first stage, maximum and minimum temperature value from thermal images processed by morphological operations are extracted. In the second stage, interval features are extracted and continuous features are produced. In the third stage, Fisher's criterion is used to transform the continuous features to new feature space which produce the input data to the classification process. They used a leave-one-out cross validation method during the training process. They reached sensitivity = 85.7% and specificity = 86.5%. De Santana et al [22] study the performance of several classification techniques and group the thermal images into one of the following groups: benign lesion, malignant lesion and cyst with the use of Haralick and Zernike descriptors for attributes extraction. They use Artificial neural networks (ANN), Multi-layer perceptron (MLP), Extreme learning machines(ELM), decision trees (DT) and Bayesian classifiers to perform the classification. They achieved accuracy of 76.01% by using MLP as classifier with 10-fold cross validation. Milosevic et al. [23] extracted 20 Gray Level Co-occurrence Matrices (GLCM) features from 40 thermal images. They used Support Vector Machine (SVM), K-Nearest Neighbor (kNN) and Naïve Bayes (NN) as Classifiers. Also, they used K-fold cross validation method with K = 5 and achieved accuracy = 92.5% by using kNN classifier and Sensitivity = 85.7% by using SVM and Naïve Bayes as classifier. The proposed system extracted the breast area manual and results can't be generalized due to limitations of the dataset. Dey et al. [24] extract 112 features by using texture features and entropy features. They used DT, KNN, SVM1 and SVM-RBF (SVM2) as classifiers. The proposed system attained an overall accuracy>89%. But, the breast area is extracted manually and a limited number of dataset is used to evaluate the proposed system Francis et al. [25] presented a curvelet transform-based feature extraction approach to detect breast abnormality from thermal images. The curvelet transform enhances the accuracy of the classification process by representing edges and distinctiveness in curves in an image. They obtained accuracy = 90.91%, Sensitivity = 81.82% and Specificity = 100% by using SVM as classifier. Pramanik et al. [26] calculated discrete wavelet transform to determine the initial feature point image of each thermal image. They used Principal Component Analysis (PCA) to reduce feature matrix dimension. Also, they used a feed-forward Perceptron on 306 thermal images and achieved accuracy = 90.48%, sensitivity = 87.6% and specificity = 89.73%. Rajinikanth et al. [27] proposed an automated breast cancer detection system from thermal images. They used two feature extraction pipelines (1) saliency enhancement, morphological segmentation and GLCM

feature mining and, (2) Local-Binary-Pattern (LBP) enhancement and feature extraction. Then, serial feature integrations are implemented and Marine-Predators-Algorithm (MPA) is used to choose the optimized features. The optimized features are used to evaluate the performance of different SVM classifiers. They achieved accuracy of 93.5 by using SVM cubic (SVM-C) and SVM Coarse Gaussian (SVM-CG).

Deep learning approaches have recently been developed to extract characteristics and improve the efficiency of medical image analysis. Deep learning is one of machine learning methods, which uses multilayer convolutional neural networks (CNN). Unlike other feature extraction techniques, the CNN is able to extract the features of the images from the dataset directly. This type of feature extraction is used to extract features from different parts of the image using convolution. Mambou et al. [28] proposed a deep neural network (DNN) model depending on a pre- trained Inception V3 model [29] for the classification of sick breast and healthy breast. They involved SVM classifier in the case of an uncertainty in the output of the DNN; additionally, they catch attention to the breast's physical model camera sensitivity. Gomaz et al. [17] study the impact of data preprocessing, data augmentation and the size of database versus a proposed set of CNN models. Also, they used a tree Parzen estimator to develop a CNN hyper-parameters fine- tuning optimization model. They achieved an accuracy of 92% and F1-score of 92%. Cabıoğlu and Oğul [30] designed various CNNs by using transfer learning technique. They achieved an accuracy of 94.3%, a precision of 94.7% and a recall of 93.3%. But, they didn't use a segmentation method to extract the breast area from other parts of the thermal images. Barbosa et al. [31] used deep-wavelet neural networks(DWNN) as a feature extraction technique. They found that when features' number increases, by adding additional levels in the DWNN, better performance can be achieved in solving the classification problem. They obtained 95% of sensitivity and 79% of specificity. Based on bio-data, image analysis, and image statistics. Ekici and Jawzal [32] suggested a new technique for feature extraction. To classify the breast images as normal or suspicious, they used a CNN optimized by the Bayes algorithm. They achieved accuracy around 99%.

From the discussed related work above, it could be remarked that the prior work has some limitations such as:

1. some related work used a small number of the dataset as in [8, 23].

2. some related work did not consider segmentation of the breast area before classification such as in [30] or extract the breast area manually such as in [23, 24].

3. some segmentation models such as in [1] removed parts of the breast.

4. some work has been evaluated by only calculating the accuracy metric only such as in [32]. However, the high accuracy rate of a model does not ensure its ability to distinguish different classes equally if the dataset is unbalanced [39].

Therefore, a fully automated breast cancer detection system from thermograms is needed and should be evaluated by not only the accuracy but also the most related metrics such as sensitivity and specificity.

## 3. Proposed method

To automate and improve the accuracy of thermography systems, we designed a deep learning-based system which integrates U-Net network and a proposed deep learning model. The proposed system is a combination two important methods: U-Net network and a two-class CNN-based deep learning model. First, U-Net is a convolutional network architecture which proved very strong in biomedical segmentation and very fast compared with other methods

[33]. U-Net is used in our system to automatically extract and isolate the breast area from other parts of the body which act as noise in the detection system. Second, the two-class CNN-based deep learning model is trained from scratch to extract more characteristics from the dataset that is helpful in training the network and improve the efficiency of the classification process. The novelty of the proposed system lays in using U-Net network for automating the segmentation process and building a deep learning model which use the output of U-Net to classify the given thermogram. The combination between U-Net and our proposed deep learning model proved to be effective as it achieved accuracy = 99.33%, sensitivity = 100% and specificity = 98.67%.

The proposed method is divided into three phases, image resizing, breast area segmentation and deep learning model for classification. Fig 2 summarized the proposed method in a flowchart.

### 3.1 Image resizing

The thermal images are of size 680 × 480 pixels and its computation time will be high due to the limitation of the PC capabilities used in this study. So, the thermal images are resized to a smaller size of 228 × 228 pixels for faster computation.

### 3.2 Breast area segmentation using deep learning (CNN)

The thermal image contains unnecessary areas as neck, shoulder, chess and other parts of the body which acts as noise during the training in CNN models. The aim of this phase is removing unwanted regions and using the areas destined to have cancer as the input of the CNN model for training and testing.

Because finding a large training dataset in medical problems is challenging, Ronneberger et al. proposed the U-NET network structure [33], a convolutional network architecture for biomedical segmentation that has a good influence on smaller training datasets [34]. So, we use U-net network for breast area segmentation from thermal images. U-net consists of a contracting path (left side) and an expansive path (right side). The contracting path consists of two 3x3 convolutions (unpadded convolutions) that are applied repeatedly, each followed by a rectified linear unit (ReLU) and a 2x2 max-pooling operation with stride 2 for downsampling. The number of feature channels is doubled with each downsampling step. In the expansive

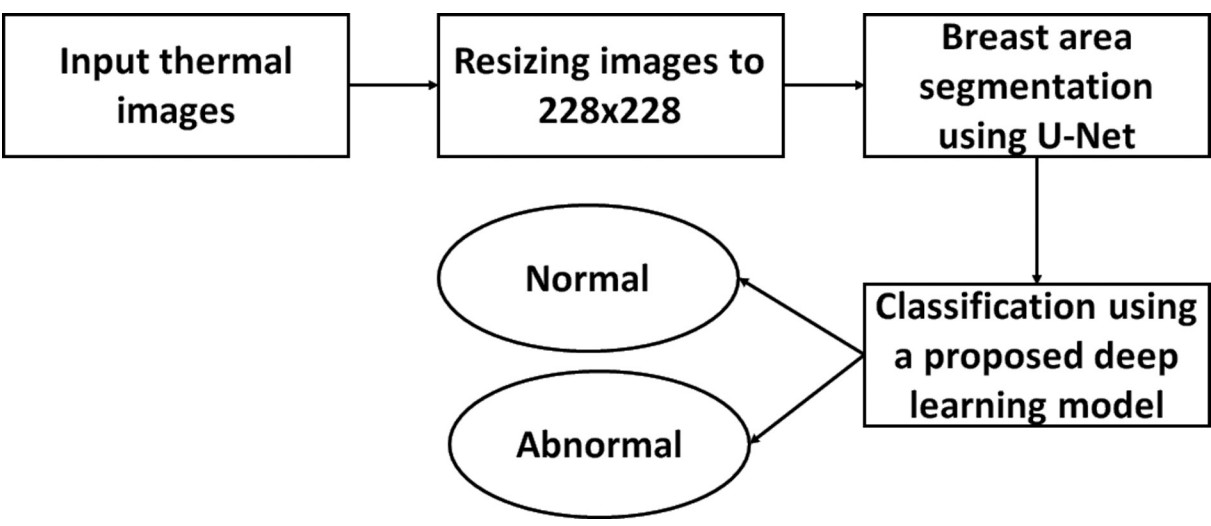

**Fig 2. Flowchart of the proposed method.**

path, an upsampling of the feature map is followed by a 2x2 convolution that halves the number of feature channels, a concatenation with the proportionally cropped feature map from the contracting path, and two 3x3 convolutions, each followed by a ReLU. Also, U-net has some advantages such as it has a simple structure, less parameters, needs very few images for training (approximately 30 per application) and the training time is relatively short compared with other networks [35]. Example of U-net architecture is shown in Fig 3. The initial input dimension in UNet is depicted as 572x572. But, we define the network, so we can change the input dimension in the input layer to the desired one.

In U-Net, to decrease the resolution of the input image, an initial set of convolutional layers are combined with max-pooling layers. Then, in sequence, a number of convolutional layers paired with upsampling operators are applied in order to increase the resolution of the input image. When these two pathways are combined, a U-shaped graph is created, which can be used to perform image segmentation. Fig 4 shows example of using U-net network for breast area segmentation.

### 3.3 Deep learning model for classification

Convolution layer, Rectified Linear Activation Function (RELU) layer, max pooling layer, fully connected layer, and dropout layer are the five parts of a CNN model. The most significant part of CNN is the convolution layer. It consists of trainable filters and updates its parameters at each iteration. RELU layer is the most preferred layer in CNN architectures as it speeds up the training process. Max pooling layer is used to reduce parameter size and control overfitting. Neurons in fully connected layer are a regular neural network. Dropout layer is used to prevent overfitting.

A two-class CNN-based deep learning model, which is trained from scratch and used to classify normal and abnormal breast tissue, is proposed. The network has nine layers, as illustrated in Fig 5, with the first six being convolutional layers and the remaining three being fully connected layers. In the proposed model, the first layer filters the input image, of size $228 \times 228$ pixels, with 64 kernels of size $7 \times 7$ with a stride of 6 pixels. Kernels of the first layer are with depth = 3, which define the number of color channels of the input thermogram image. After applying max-pooling, which is used to enhance the robustness and reduce the

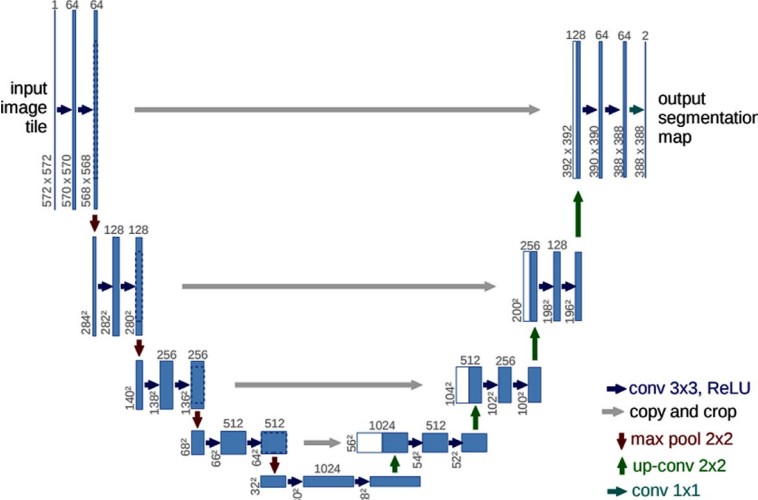

**Fig 3. Example of U-Net architecture [33].**

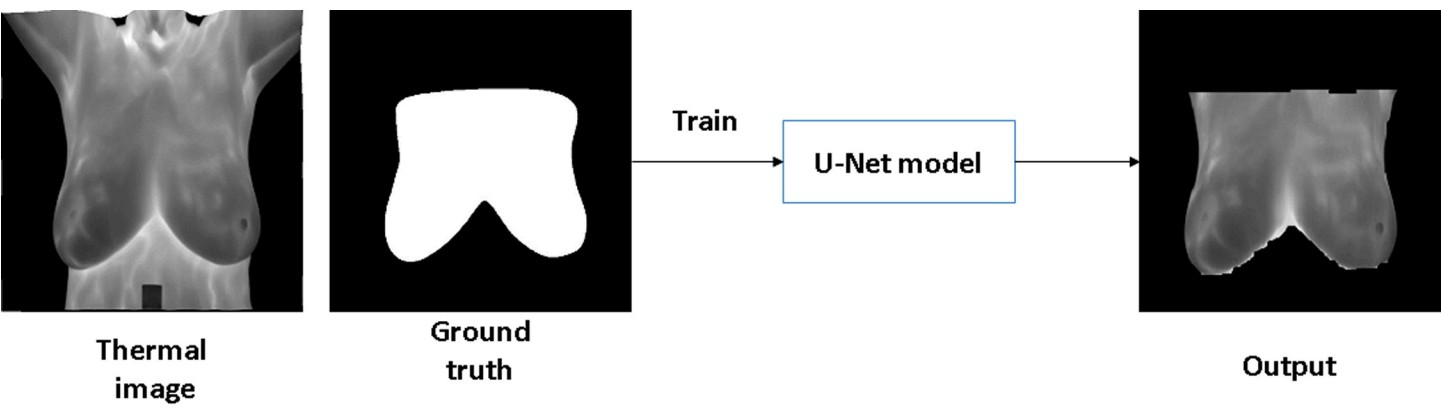

**Fig 4. Example of breast area segmentation with U-Net.**

computation, the output of the first layer is used as input for the second layer, filtering it with 128 kernels of size $3 \times 3 \times 64$. Without pooling layers, the third, fourth, and fifth levels are connected to each other. The third layer consists of 256 kernels with a size of $3 \times 3 \times 128$. The fourth layer has 256 kernels of size $3 \times 3 \times 256$ and the fifth layer consists of 256 kernels with a size of $3 \times 3 \times 256$. The sixth layer is connected to the fifth layer with max-pooling layer and has 256 kernels of size $3 \times 3 \times 256$. On top of the convolutional layers, two fully-connected layers of 1024 neurons are connected to each other. The number of neurons in the third fully-connected layer equals the number of classes. The output of the convolutional and the max-pooling layers is calculated by Eqs (1) and (2), respectively.

$$O_{conv} = \frac{I - K + 2P}{S} + 1 \tag{1}$$

$$O_{pooling} = \frac{I - P_s}{S} + 1 \tag{2}$$

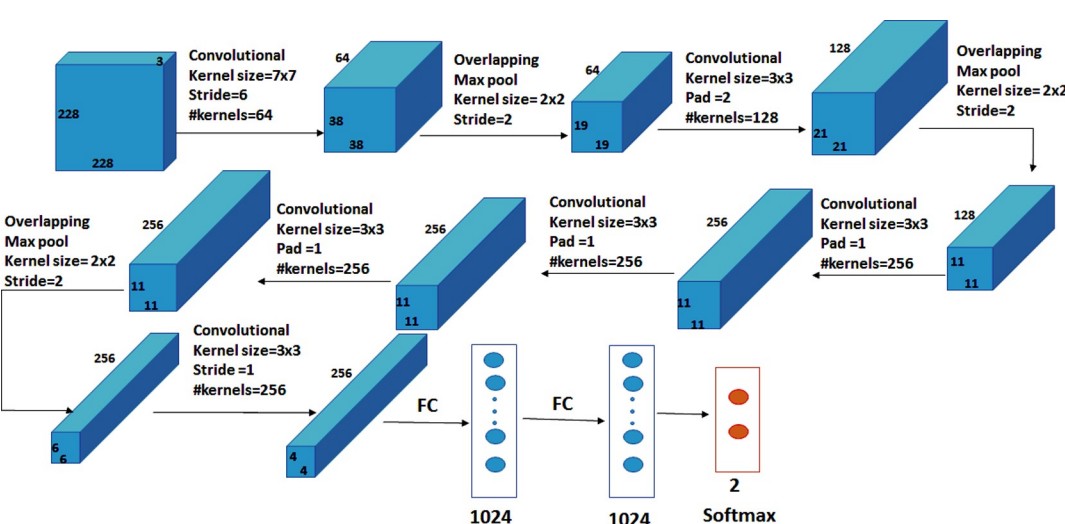

**Fig 5. Architecture of the proposed deep learning model.**

where $O_{conv}$ is the size of the output of the convolutional layer, $I$ is the size of the input layer, $K$ is the size of kernels used in the convolutional layer, $P$ is padding, $S$ is the stride of the convolution operation, $O_{pooling}$ is the size of the output of the max-pooling layer and $P_s$ is the pool size. The CNN model was implemented using the Matlab 2019b platform running on a laptop computer system with the following specifications: Intel (R) Core (TM) i7-2670 CPU@2.20GHZ with 8 GB RAM.

## 4. Experimental results

To evaluate the proposed method, a benchmark database (DMR-IR) [19] was used. This database is created by collecting the IR images from the Hospital of UFF University and published publicly with the approval of the ethics committee where consent should be signed by any patient. This study used a set of 1000 frontal thermogram images, captured using a FLIR SC-620 IR camera with a resolution of $640 \times 480$ pixels from this database (including 500 normal and 500 abnormal subjects). These images contain breasts in various shapes and sizes (see Fig 6). The dataset is split for segmentation and classification into training, validation and testing sets with the ratio 70:15:15, randomly. The dataset description is included in Table 1.

### 4.1 Breast area segmentation using deep learning (CNN)

During the training of breast segmentation phase with U-Net network, Adaptive Moment Estimation (ADAM) method was used as optimized algorithm with number of epochs = 30. Also, the training process was started with initial learning rate = 1.0e−3. The learning rate used a piecewise schedule and dropped by a factor of 0.3 every 10 epochs to allow the network to train quickly with a higher initial learning rate. The network trained with an 8-batch size to save memory. Fig 7 shows examples of breast area segmentation results.

### 4.2 Evaluation of the deep learning model

Classification Metrics evaluate the performance of the model and measure how good or bad the classification is.

**Accuracy:** Represents how many instances are completely classified correctly. It is calculated by dividing the total number of predictions by the number of right predictions. It is

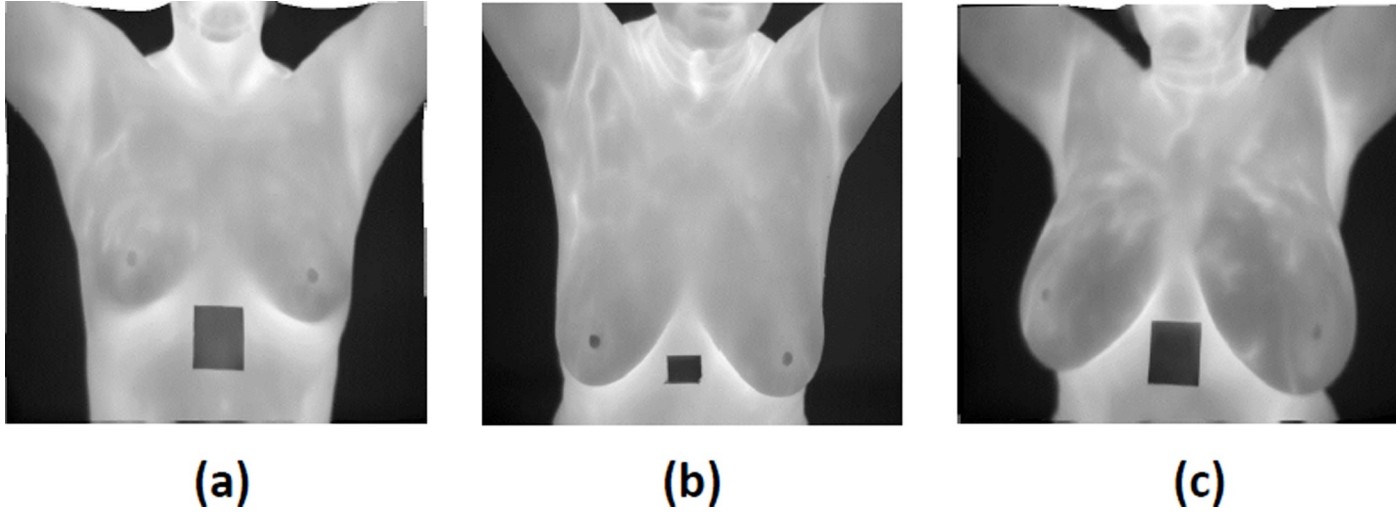

**Fig 6.** Different cases of breast (a) small breast (b) large breast (c) asymmetric breast.

**Table 1. Dataset description.**

| Dataset categories | Dimension | Training | Validation | Testing | Total |
|---|---|---|---|---|---|
| Normal | 640x480 | 350 | 75 | 75 | 500 |
| Abnormal | 640x480 | 350 | 75 | 75 | 500 |

calculated by Eq (3)

$$Accuracy = \frac{T_P + T_N}{T_P + T_N + F_P + F_N} \tag{3}$$

**Sensitivity:** Is calculated based on how many patients have the disease are correctly estimated. It is calculated by Eq (4)

$$Sensitivity = \frac{T_P}{T_P + F_N} \tag{4}$$

**Specificity:** Is calculated based on how many patients do not have the disease are predicted right. It is calculated by Eq (5)

$$Specificity = \frac{T_N}{T_N + F_P} \tag{5}$$

- *True Positive* (TP) refers to a positive-class sample that has been successfully classified by a model.

- *False Positive* (FP) refers to a sample that should have been classed as negative but was instead classified as positive.

- *True Negative* (TN) refers to a negative-class sample that has been successfully classified by a model.

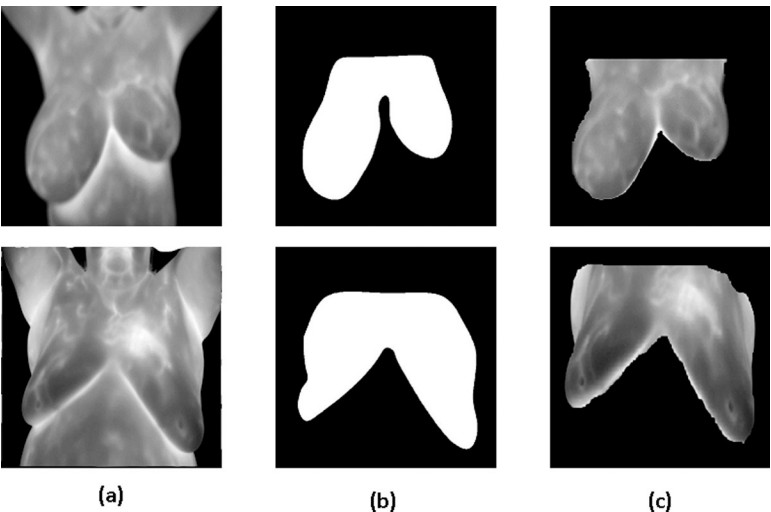

(a)　　　　　(b)　　　　　(c)

**Fig 7.** Breast area segmentation resuls (a) thermal image (b) ground truth (c) output.

- *False Negative* (FN) refers to a sample that should have been classed as positive but was instead classified as negative.

The accuracy metric indicates how many of the model's predictions were right. However, if the dataset is unbalanced, a model's high accuracy rate does not guarantee its ability to differentiate distinct classes equally. In specifically, in the classification of medical images, it is necessary to develop a model with the ability be applied to all classes. In cases, sensitivity and specificity should be used to provide information about the performance of the model. Sensitivity measures [3] the percentage of patient have the disease that the proposed model correctly predicted. Specificity measures the percentage of patient do not have the disease and correctly estimated by the proposed model. These two evaluation metrics measure the ability of the model to decrease *FN* and *FP* predictions.

In the training process, we use Adaptive Moment Estimation (ADAM)method as solver with batch size of 60 and number of epochs = 30. Also, the training process was started with initial learning rate = 2.0e−3. According to the training parameters, we achieve accuracy = 99.33%, sensitivity = 100% and specificity = 98.67%. The training progress and the confusion matrix of the proposed model are shown in Figs 8 and 9, respectively.

## 4.3 Impact of changing the training options on the classification process

We further study the impact of the training options on the classification accuracy, sensitivity and specificity. In Table 2, we show the influence of three different solvers, Stochastic Gradient Descent with Momentum(SGDM) [36], Adaptive Moment Estimation (ADAM) [37] and Root Mean Square propagation(RMSprop) [38]. In this table, the training process was started with initial learn rate = 2.0e−3, batch size was = 60 and number of epochs was = 30. The impact of starting the training process with different number of epochs is shown in Table 3. In Table 3, ADAM was used as solver with batch size of 60 and the initial learn rate of the training process was 2.0e−3. In Table 4, we show the impact of using different batch size, in the training process, on the classification accuracy, sensitivity and specificity. In this table, ADAM was used as solver with number of epochs = 30 and the initial learn rate was 2.0e−3. In Table 5, we show the impact of starting the training process with 32 different initial learning rates on the

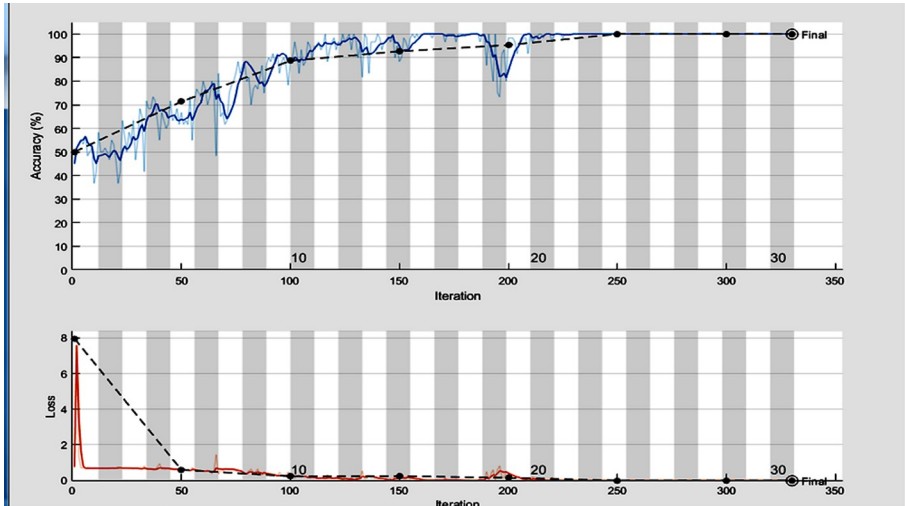

**Fig 8. The training progress of the proposed deep learning model.**

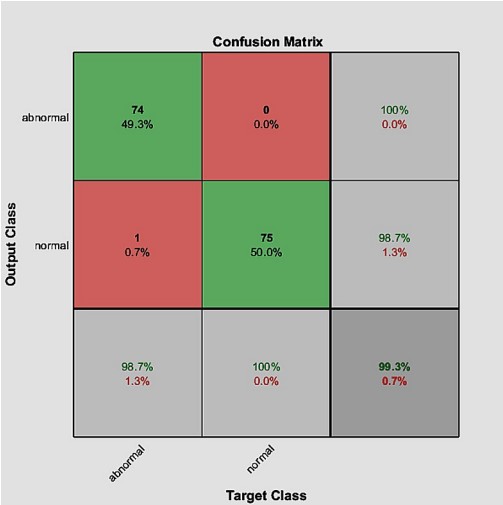

**Fig 9. The confusion matrix of the proposed model.**

classification accuracy, sensitivity and specificity. In this table, ADAM was used as solver with batch size = 60 and number of epochs = 30

## 4.4 Performance of pretrained CNN models on the dataset

The performance of different pretrained CNN models such as ResNet18, GoogleNet [22], VGG16 and AlexNet is performed on the same Dataset [39]. A comparison between the evaluation metrics of pretrained CNN models and the proposed model is shown in Table 6. According to results in Table 6, we note that the performance of the proposed model is better than the performance of other CNN models on this dataset except VGG16Net.

## 4.5 Impact of training/testing data size

The impact of dataset size is measured on the performance of the proposed model. Fig 6 plots the performance of three evaluation metrics on different dataset size. In this part of the experiment, ADAM was used as solver, number of epochs = 30, batch size = 60 and validation data = 15%. Fig 10 shows that the dataset size is critical on the classification process.

## 4.6 Performance of machine learning classifier on the dataset

We further study the performance of machine learning classifier such as SVM, KNN and Decision Tree on the classification process on the same dataset. First, we extract texture features by using Gray Level Co-occurrence Matrices (GLCM) [40] and Histogram of Oriented Gradients (HOG) [41]. Then, we distinguish between normal and abnormal breast tissue by using the classifiers. A comparison between the performance of machine learning classifiers with GLCM and HOG features extraction methods and the proposed model are shown in Tables 7 and 8.

**Table 2. Comparison between solvers (initial learn rate = 2.0e−3, number of epochs = 30 and batch size = 60).**

| Solver | Accuracy (%) | Sensitivity (%) | Specificity (%) |
|:---:|:---:|:---:|:---:|
| ADAM | 99.33 | 100 | 98.67 |
| SGDM | 84.17 | 100 | 68.33 |
| RMSprop | 50 | 100 | 0.0 |

**Table 3. The impact of using different number of epochs on the classification accuracy, sensitivity and specificity (solver = ADAM, initial learn rate = 2.0e−3, batch size = 60).**

| Number of Epochs | Accuracy (%) | Sensitivity (%) | Specificity (%) |
|:---:|:---:|:---:|:---:|
| 10 | 88.67 | 94.67 | 82.67 |
| 20 | 97.33 | 100 | 94.67 |
| 30 | 99.33 | 100 | 98.67 |
| 40 | 100 | 100 | 100 |
| 50 | 100 | 100 | 100 |

### 4.7 Statistical analysis

To analyze the evaluation of the proposed system statistically, we perform the analysis of variance (ANOVA) test [18, 42], where the proposed system is compared with ResNet18, GoogleNet and VGG16 networks. The result of the ANOVA test is shown in Table 9. To reject the null hypothesis, the *p-value* in the ANOVA test should be less than 0.05. According to Table 9, the *p−value* is less than 0.05. so, the null hypothesis was rejected by the results of ANOVA test.

## 5. Discussion

In this study, we propose a fully automatic breast cancer detection system. The proposed system uses U-Net network to extract the breast area from thermal images and propose a deep learning model, which is trained for the classification of abnormal breast tissues using thermal images. The proposed system consists of three main phases, resizing, breast area segmentation and deep learning model for classification. In resizing phase, the thermal images are resized to a smaller size to accelerate computation. In breast area segmentation phase, the breast region is extracted automatically by using U-Net network. In deep learning model for classification phase, we proposed a two-class CNN-based deep learning model, which is trained from scratch and used for the classification of normal and abnormal breast detection.

The experimental results obtained show an overview of our contribution in (1) extracting the breast area from the thermal images automatically (2) studying the impact of the training options on the classification accuracy, sensitivity and specificity. (3) comparing between the performance of pretrained CNN models such as ResNet18, GoogleNet, AlexNet, VGG16Net and the proposed model. (4) comparing between machine learning classifier such as SVM, KNN and Decision Tree and the proposed model. In Table 2, we study the influence of three different solvers, SGDM, ADAM and RMSprop on the classification process. From Table 2, we

**Table 4. Impact of using different batch size on the classification accuracy, sensitivity and specificity (solver = ADAM, initial learn rate = 2.0e−3, number of epochs = 30).**

| Batch Size | Accuracy (%) | Sensitivity (%) | Specificity(%) |
|:---:|:---:|:---:|:---:|
| 10 | 50 | 100 | 0.0 |
| 20 | 100 | 100 | 100 |
| 30 | 100 | 100 | 100 |
| 40 | 100 | 100 | 100 |
| 50 | 100 | 100 | 100 |
| 60 | 99.33 | 100 | 98.67 |
| 70 | 98.67 | 100 | 97.33 |
| 80 | 95.83 | 100 | 91.67 |
| 90 | 91.33 | 100 | 82.66 |
| 100 | 83.33 | 100 | 66.67 |

**Table 5. Impact of starting the training process with different initial learn rate on the classification accuracy, sensitivity and specificity (solver = ADAM, batch size = 60 and number of epochs = 30).**

| Initial learning rate | Accuracy(%) | Sensitivity(%) | specificity(%) |
|---|---|---|---|
| $9.0\,e^{-01}$ | 50 | 100 | 0.0 |
| $8.0\,e^{-01}$ | 50 | 100 | 0.0 |
| $7.0\,e^{-01}$ | 50 | 100 | 0.0 |
| $6.0\,e^{-01}$ | 50 | 100 | 0.0 |
| $5.0\,e^{-01}$ | 50 | 100 | 0.0 |
| $4.0\,e^{-01}$ | 50 | 100 | 0.0 |
| $3.0\,e^{-01}$ | 50 | 100 | 0.0 |
| $2.0\,e^{-01}$ | 50 | 100 | 0.0 |
| $1.0\,e^{-01}$ | 50 | 100 | 0.0 |
| $9.0\,e^{-02}$ | 50 | 100 | 0.0 |
| $8.0\,e^{-02}$ | 50 | 100 | 0.0 |
| $7.0\,e^{-02}$ | 50 | 100 | 0.0 |
| $6.0\,e^{-02}$ | 50 | 100 | 0.0 |
| $5.0\,e^{-02}$ | 50 | 100 | 0.0 |
| $4.0\,e^{-02}$ | 50 | 100 | 0.0 |
| $3.0\,e^{-02}$ | 50 | 100 | 0.0 |
| $2.0\,e^{-02}$ | 50 | 100 | 0.0 |
| $1.0\,e^{-02}$ | 50 | 100 | 0.0 |
| $9.0\,e^{-03}$ | 50 | 0.0 | 100 |
| $8.0\,e^{-03}$ | 50 | 0.0 | 100 |
| $7.0\,e^{-03}$ | 83.33 | 100 | 66.67 |
| $6.0\,e^{-03}$ | 41.07 | 10.0 | 73.33 |
| $5.0\,e^{-03}$ | 51.67 | 5.0 | 98.33 |
| $4.0\,e^{-03}$ | 56.67 | 13.33 | 100 |
| $3.0\,e^{-03}$ | 83.33 | 100 | 66.67 |
| $2.0\,e^{-03}$ | 99.33 | 100 | 98.67 |
| $1.0\,e^{-03}$ | 99.33 | 100 | 98.67 |
| $9.0\,e^{-04}$ | 100 | 100 | 100 |
| $8.0\,e^{-04}$ | 100 | 100 | 100 |
| $7.0\,e^{-04}$ | 100 | 100 | 100 |
| $6.0\,e^{-04}$ | 100 | 100 | 100 |
| $5.0\,e^{-04}$ | 100 | 100 | 100 |

can note that ADAM has the best behavior. In Table 3, we study the impact of starting the training process with different number of epochs. From Table 3, we can obtain that when the number of epochs is increased, accuracy, sensitivity and specificity values are increased. We further study the impact of using different batch size, in the training process, on the

**Table 6. Comparison between the performance metrics of different CNN models and the proposed model.**

| CNN model | Accuracy | Sensitivity | Specificity |
|---|---|---|---|
| ResNet18 | 93.3 | 88.0 | 98.7 |
| GoogleNet | 79.33 | 84.00 | 74.67 |
| AlexNet | 50.0 | 0.0 | 100 |
| VGG16 | 100 | 100 | 100 |
| Proposed CNN | 99.33 | 100 | 98.67 |

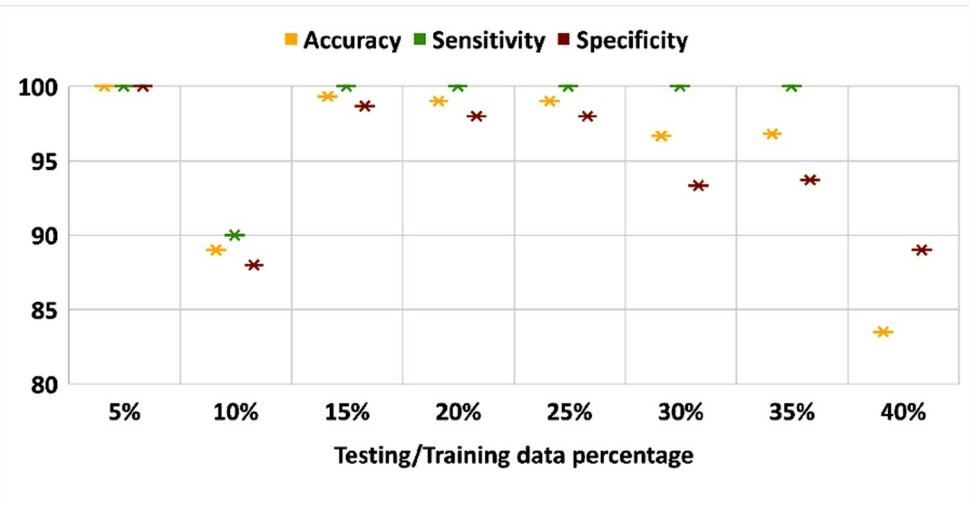

**Fig 10. Evaluation metrics over different dataset size.**

classification process in Table 4. From this table, we can obtain that when the batch size = 10 the behavior of the classification process is very bad and when the batch size value is from 20 to 50, accuracy, sensitivity and specificity have constant values. Also, when the batch size is greater than 50, the accuracy and specificity values are decreased but the sensitivity has a constant value. In Table 5, we show the impact of starting the training process with different initial learning rates on the classification process. From Table 5, it is obtained that when the learning rate is greater than 3.0 $e^{-03}$, the behavior of the classification process becomes very bad except at learning rate = 7.0 $e^{-03}$. In Table 6, we perform a comparison between the performance of pretrained CNN models such as ResNet18, GoogleNet, AlexNet, VGG16Net and the proposed model. From Table 6, we can note that the performance of the proposed model is better than the performance of other CNN models on this dataset except VGG16Net. In Fig 6, the impact of dataset size is measured on the performance of the proposed model. From Fig 6, we can note that when the training data size is increased and the testing data size is decreased the accuracy values of the proposed model is increased except when the testing data size = 10%. In addition, we compare between the performance of machine learning classifiers with GLCM and HOG features extraction methods and the proposed model in Tables 7 and 8 and we can note that our proposed model has the best result of the performance metrics. From Table 8, we can note that KNN and Decision Tree classifiers have a very bad results on this dataset with HOG features.

To further evaluate our proposed system, as shown in Table 10, a comparison between the proposed system and other studies based on breast area segmentation and breast cancer detection is performed. From this table, we can note that the dataset used by our proposed system is large compared with the dataset of some related work. Also, our system extracts the breast area

**Table 7. Comparison between the performance metrics of different machine learning classifier with texture features and the proposed model.**

| Classifier | Accuracy(%) | Sensitivity(%) | specificity(%) |
|---|---|---|---|
| SVM | 89.33 | 86.67 | 92.0 |
| KNN | 53.33 | 64.0 | 42.67 |
| Decision Tree | 82.67 | 96.00 | 96.0 |
| Proposed method | 99.33 | 100 | 98.67 |

**Table 8. Comparison between the performance metrics of different machine learning classifier with HOG features and the proposed model.**

| Classifier | Accuracy(%) | Sensitivity(%) | specificity(%) |
|---|---|---|---|
| SVM | 78.28 | 73.33 | 86.67 |
| KNN | 20.0 | 20.0 | 20.0 |
| Decision Tree | 40.0 | 58.67 | 21.33 |
| Proposed method | 99.33 | 100 | 98.67 |

**Table 9. Results of the ANOVA test of the proposed model and CNN models.**

| Model | P-value |
|---|---|
| ResNet18 | 0.0423 |
| GoogleNet | 0.0173 |
| VGG16 | 0.0023 |

**Table 10. Comparison with other studies on breast cancer detection (n = normal, ab = abnormal, Ea = Early, Ac = Acute).**

| Ref. | Segmentation method | #patients / Thermograms | Classification Method | Results |
|---|---|---|---|---|
| [8] | an enhanced segmentation method based on both Neutrosophic sets (NS) and optimized Fast Fuzzy c-mean (F-FCM) algorithm. | 63 thermograms (29 N / 34 AB) | SVM Classifier | Accuracy = 92.06% |
| | | | | Precision = 87.5% |
| | | | | Recall = 96.55% |
| [23] | Manual | 40 thermograms (26 N / 14 AB) | SVM, Naïve Bayes and KNN classifier | Accuracy = 92.5% and Sensitivity = 78.6% with KNN |
| | | | | Accuracy = 85% and Sensitivity = 85.7% with SVM |
| | | | | Accuracy = 80% and Sensitivity = 85.7% with Naïve Bayes |
| [24] | Manual | 68 thermograms (26 Ea / 42 Ac) | DT, KNN, SVM and SVM-RBF | Accuracy = 95.59%, |
| | | | | Sensitivity = 96% and Specificity = 95.35% with SVM-RBF |
| [25] | Canny edge detection methods followed by gradient operators and Hough transform for boundary detection | Thermograms of 22 women (11 N / 11AB) | SVM Classifier | Accuracy = 90.91%, |
| | | | | Sensitivity = 81.82% |
| | | | | Specificity = 100% |
| [26] | Otsu's threshold to remove background followed by a reconstruction technique. | 306 thermograms (183 N / 123 AB) | Feed-forward artificial neural network with gradient decent | Accuracy = 90.48%, |
| | | | | Sensitivity = 87.6%, |
| | | | | Specificity = 89.73% |
| [27] | Manual | 600 thermograms (300 N / 300 AB) | SVM-C | Accuracy = 93.5%, |
| | | | | Sensitivity = 93%, |
| | | | | Specificity = 94% |
| [30] | Not defined | 282 thermograms (147 N / 135 AB) | CNN using transfer learning | Accuracy = 94.3% |
| | | | | Precision = 94.7% |
| | | | | Recall = 93.3% |
| [32] | Projection profile analysis | 140 patients (98 N / 32 AB) | Convolutional Neural Networks optimized by Bayes algorithm | Accuracy = 98.95% |
| Proposed method | U-Net network | 1000 thermograms (500 N / 500 AB) | Two-class CNN-based deep learning model | Accuracy = 99.33%, |
| | | | | Sensitivity = 100%, |
| | | | | Specificity = 98.67% |

automatically by using U-Net network, but some related work doesn't used segmentation method and other extract it manually. In addition, the evaluation metric of our proposed system is better than related work. So, the proposed system outperformed other models. Furthermore, Statistical analysis by ANOVA test indicates the viability of the proposed system. In addition, the proposed system is domain-independent, so it has the ability to be applied to various computer vision tasks.

It is worth mention that the study has some limitations: the computation time of the segmentation process is high due to the limitation of the PC capabilities used in this study as well as the proposed deep learning model for classification has a bad behavior when the learning rate is greater than 3.0 e−03

## 6. Conclusion

Breast cancer is one of the most commonly diagnosed malignancies in women around the world. Several researches have worked on breast cancer segmentation and classification using variety of imaging techniques. Thermography imaging is an effective diagnostic approach which is used for breast cancer detection with the help of infrared technology. In this paper, we propose a fully automatic breast cancer detection system. The proposed method is divided on three main stages. First, the thermal images are resized to a smaller size to accelerate computation. Second, the breast region is extracted automatically by using U-Net network. Third,

**Table 11.  Table of abbreviation.**

| Abbreviation | Definition |
|---|---|
| CNN | Convolutional Neural Networks |
| CAD | Computer-Aided Detection |
| ROI | Region of Interest |
| EHMM | Extended Hidden Markov Models |
| NS | Neutrosophic Sets |
| F-FCM | Fast Fuzzy C-Mean |
| SVM | Support Vector Machine |
| KNN | K-Nearest Neighbor |
| NN | Naïve Bayes |
| DT | Decision Tree |
| PCA | Principal Component Analysis |
| DNN | Deep Neural Network |
| DWNN | Deep-Wavelet Neural Networks |
| RELU | Rectified Linear Activation Function |
| ADAM | Adaptive Moment Estimation |
| SGDM | Stochastic Gradient Descent with Momentum |
| RMSprop | Root Mean Square propagation |
| $T_P$ | True Positive |
| $T_N$ | True Negative |
| $F_P$ | False Positive |
| $F_N$ | False Negative |
| GLCM | Gray Level Co-occurrence Matrices |
| HOG | Histogram of Oriented Gradients |
| n | normal |
| ab | abnormal |

a deep learning model based two-class CNN is proposed and trained from scratch for the classification of normal and abnormal breast tissue.

Based on the experimental results, the proposed model achieved accuracy = 99.33%, sensitivity = 100% and specificity = 98.67%. In Table 10, a comparison between the proposed system and other studies based on breast area segmentation and breast cancer detection is performed. Furthermore, Statistical analysis by ANOVA test indicates the viability of the proposed system. In addition, the proposed system is domain-independent, so it has the ability to be applied to various computer vision tasks. In future study, we will investigate deep learning models which can highlight and label defect region using thermal images.

## List of abbreviations

Table 11 presents the definition of the abbreviations used in this paper.

## Supporting information

**S1 Table. Evaluation metrics of the proposed system over different dataset sizes.** (XLSX)

## Acknowledgments

The authors would like to thank the Department of Computer Science and the Hospital of the Federal University Fluminense, Niterói, Brazil, for providing DMR-IR benchmark database which is accessible through an online user-friendly interface (*http://visual.ic.uff.br/dmi*) and used for experiments.

## Author Contributions

**Conceptualization:** Esraa A. Mohamed, Essam A. Rashed.

**Data curation:** Esraa A. Mohamed, Essam A. Rashed, Tarek Gaber.

**Formal analysis:** Esraa A. Mohamed.

**Investigation:** Esraa A. Mohamed, Tarek Gaber.

**Methodology:** Esraa A. Mohamed, Essam A. Rashed, Tarek Gaber.

**Project administration:** Esraa A. Mohamed, Essam A. Rashed, Tarek Gaber, Omar Karam.

**Resources:** Esraa A. Mohamed.

**Software:** Esraa A. Mohamed.

**Supervision:** Essam A. Rashed, Tarek Gaber, Omar Karam.

**Validation:** Essam A. Rashed, Tarek Gaber.

**Visualization:** Esraa A. Mohamed, Tarek Gaber.

**Writing – original draft:** Esraa A. Mohamed.

**Writing – review & editing:** Essam A. Rashed, Tarek Gaber, Omar Karam.

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
