## [Decision Letter · Decision Letter 0]

29 Sep 2021

PONE-D-21-28729Deep Learning Model for Fully Automated Detection of Abnormal Breast Tissues from ThermogramsPLOS ONE

Dear Dr. A. Mohamed,

Thank you for submitting your manuscript to PLOS ONE. After careful consideration, we feel that it has merit but does not fully meet PLOS ONE’s publication criteria as it currently stands. Therefore, we invite you to submit a revised version of the manuscript that addresses the points raised during the review process.

Specifically, we authors need to improve the description of their methodology and the presentation of the results, which must include statistical analysis.

We look forward to receiving your revised manuscript.

Kind regards,

Robertas Damaševičius

Academic Editor

PLOS ONE

Journal Requirements:

3. We note that Figure 1 in your submission contain copyrighted images. All PLOS content is published under the Creative Commons Attribution License (CC BY 4.0), which means that the manuscript, images, and Supporting Information files will be freely available online, and any third party is permitted to access, download, copy, distribute, and use these materials in any way, even commercially, with proper attribution. For more information, see our copyright guidelines: http://journals.plos.org/plosone/s/licenses-and-copyright.

Additional Editor Comments:

Revise the article following the reviewer comments.

Reviewers' comments:

Reviewer's Responses to Questions

**Comments to the Author**

1. Is the manuscript technically sound, and do the data support the conclusions?

Reviewer #1: Partly

Reviewer #2: Yes

2. Has the statistical analysis been performed appropriately and rigorously? 

Reviewer #1: No

Reviewer #2: Yes

3. Have the authors made all data underlying the findings in their manuscript fully available?

Reviewer #1: Yes

Reviewer #2: No

4. Is the manuscript presented in an intelligible fashion and written in standard English?

Reviewer #1: Yes

Reviewer #2: Yes

5. Review Comments to the Author

Reviewer #1: The manuscript sounds technically poor, I have following concerns should be addressed before any decision. The paper currently need revision.

1. The existing literature should be classified and systematically reviewed, instead of being independently introduced one-by-one.

2. The abstract is too general and not prepared objectively. It should briefly highlight the paper's novelty as what is the main problem, how has it been resolved and where the novelty lies?

3. The 'conclusions' are a key component of the paper. It should complement the 'abstract' and normally used by experts to value the paper's engineering content. In general, it should sum up the most important outcomes of the paper. It should simply provide critical facts and figures achieved in this paper for supporting the claims.

4. For better readability, the authors may expand the abbreviations at every first occurrence.

5. The author should provide only relevant information related to this paper and reserve more space for the proposed framework.

6. However, the author should compare the proposed algorithm with other recent works or provide a discussion. Otherwise, it's hard for the reader to identify the novelty and contribution of this work.

7. The descriptions given in this proposed scheme are not sufficient that this manuscript only adopted a variety of existing methods to complete the experiment where there are no strong hypothesis and methodical theoretical arguments. Therefore, the reviewer considers that this paper needs more works.

8. Key contribution and novelty has not been detailed in manuscript. Please include it in the introduction section

9. What are the limitations of the related works

10. Are there any limitations of this carried out study?

11. How to select and optimize the user-defined parameters in the proposed model?

12. There are quite a few abbreviations are used in the manuscript. It is suggested to use a table to host all the frequently used abbreviations with their descriptions to improve the readability

13. Explain the evaluation metrics and justify why those evaluation metrics are used?

14. Some sentences are too long to follow, it is suggested that to break them down into short but meaningful ones to make the manuscript readable.

15. The title is pretty deceptive and does not address the problem completely.

16. Every time a method/formula is used for something, it needs to be justified by either (a) prior work showing the superiority of this method, or (b) by your experiments showing its advantage over prior work methods - comparison is needed, or (c) formal proof of optimality. Please consider more prior works.

17. The data is not described. Proper data description should contain the number of data items, number of parameters, distribution analysis of parameters, and of the target parameter itself for classification.

18. The related works section is very short and no benefits from it. I suggest increasing the number of studies and add a new discussion there to show the advantage. Following studies can be considered

a. Breast-Cancer Detection using Thermal Images with Marine-Predators-Algorithm Selected Features.

b. Dilated Semantic Segmentation for Breast Ultrasonic Lesion Detection Using Parallel Feature Fusion.

c. Machine-Learning-Scheme to Detect Choroidal-Neovascularization in Retinal OCT Image

d.

19. Use Anova test to record the significant difference between performance of the proposed and existing methods.

Reviewer #2: Dear Authors,

This work employs U-Net model to extract and evaluate the abnormal section from the Breast Thermal Images.

I request you to consider the following suggestions:

1. The image size is fixed as 228x228, but, the initial input dimension in UNet is depicted as 572x572. Please check and correct.

2. The quality of the image is to be improved in every image cases.

3. What is the need for extracting the breast section, please justify. Further, include, how the CNN scheme is trained to segment the image (a clear discription is needed).

4. The image dimension, such as test image, Ground truth and segmented image is not looks like 228x228. Further in this database, the Ground Truth is not available. Please discuss, how the GT is generated (Hope, the test images are from :http://visual.ic.uff.br/dmi/).

5. Please improve the reference section by considering few more related works.

(Please refer, the following research works:

a. Breast-Cancer Detection using Thermal Images with Marine-Predators-Algorithm Selected Features

b. An Examination System to Classify the Breast Thermal Images into Early/Acute DCIS Class

c. A hybrid framework to evaluate breast abnormality using infrared thermal images

6. I request you to compare the outcome of the proposed scheme with another results existing in the literature.

6. PLOS authors have the option to publish the peer review history of their article (what does this mean?). If published, this will include your full peer review and any attached files.

Reviewer #1: No

Reviewer #2: No

---

## [Author Response · Author response to Decision Letter 0]

12 Nov 2021

Original Manuscript ID: PONE-D-21-28729 

Original Article Title: “Deep Learning Model for Fully Automated Detection of Abnormal Breast Tissues from Thermograms “

Dear Editor,

Thank you for allowing a resubmission of a revised version of the manuscript, with an opportunity to address the reviewers’ comments.

We are uploading (a) our point-by-point response to the comments (below) (response to reviewers), (b) an updated manuscript with yellow highlighting indicating changes, and (c) a clean updated manuscript without highlights 

Best regards,

Authors, 

Reviewer#1, Concern # 1: The existing literature should be classified and systematically reviewed, instead of being independently introduced one-by-one.

The literature is classified systematically as recommended

Reviewer#1, Concern # 2: The abstract is too general and not prepared objectively. It should briefly highlight the paper's novelty as what is the main problem, how has it been resolved and where the novelty lies?

The abstract is edited as recommended.

Reviewer#1, Concern # 3: The 'conclusions' are a key component of the paper. It should complement the 'abstract' and normally used by experts to value the paper's engineering content. In general, it should sum up the most important outcomes of the paper. It should simply provide critical facts and figures achieved in this paper for supporting the claims.

A conclusion section is added to the manuscript with your instructions.

Reviewer#1, Concern # 4: For better readability, the authors may expand the abbreviations at every first occurrence.

 The manuscript is edited with abbreviation definition at every first occurrence.

Reviewer#1, Concern # 5: The author should provide only relevant information related to this paper and reserve more space for the proposed framework.

Information that is not related to this paper were removed. See the update in the Revised Manuscript with Track Changes page 16.

Reviewer#1, Concern # 6: However, the author should compare the proposed algorithm with other recent works or provide a discussion. Otherwise, it's hard for the reader to identify the novelty and contribution of this work.

A comparison between the proposed algorithm and other works is already exist in table 10. Also, we update this table with recent works. See the update in the Revised Manuscript with Track Changes page 18-19. From this table, we can note that the dataset used by our proposed system is large compared with the dataset of some related work. Also, our system extracts the breast area automatically by using U-Net network, but some related work doesn't used segmentation method and other extract it manually. In addition, the evaluation metric of our proposed system is better than related work. Furthermore, Statistical analysis by ANOVA test (see table 9) indicates the viability of the proposed system. In addition, the proposed system is domain-independent, so it has the ability to be applied to various computer vision tasks.

Reviewer#1, Concern # 7: The descriptions given in this proposed scheme are not sufficient that this manuscript only adopted a variety of existing methods to complete the experiment where there are no strong hypothesis and methodical theoretical arguments. Therefore, the reviewer considers that this paper needs more works.

To automate and improve the accuracy of thermography systems, we designed a deep learning-based system which integrates U-Net network and a proposed deep learning model. The proposed system is a combination two important methods: U-Net network and a two-class CNN-based deep learning model. First, U-Net is a convolutional network architecture which proved very strong in biomedical segmentation and very fast compared with other methods [29]. U-Net is used in our system to automatically extract and isolate the breast area from other parts of the body which act as noise in the detection system. Second, the two-class CNN-based deep learning model is trained from scratch to extract more characteristics from the dataset that is helpful in training the network and improve the efficiency of the classification process. The novelty of the proposed system lays in using U-Net network for automating the segmentation process and building a deep learning model which use the output of U-Net to classify the given thermogram. The combination between U-Net and our proposed deep learning model proved to be effective as it achieved accuracy= 99.33%, sensitivity=100% and specificity=98.67% 

Reviewer#1, Concern # 8: Key contribution and novelty has not been detailed in manuscript. Please include it in the introduction section.

We add the main contribution of this paper in the introduction section. To quickly and easily check this update, it is in page 4 of Revised Manuscript with Track Changes, it is also copied below:

The main contribution of this paper is as following:

1-Extracting and isolating the breast area automatically from other parts of thermal images by using CNN (U-Net).

2- Proposing a deep learning model for the classification of normal and abnormal breast tissues from thermograms

3-Evaluating the performance of the proposed model using accuracy, sensitivity and specificity.

4-Comparing the proposed model with state-of art methods.

Reviewer#1, Concern # 9: What are the limitations of the related works?

Limitations of the related work are added in the literature review section.

Reviewer#1, Concern # 10: Are there any limitations of this carried out study?

 The computation time of the segmentation process is high due to the limitation of the PC capabilities used in this study.

 The proposed deep learning model for classification has a bad behavior when the learning rate is greater than 3.0 e−03

Reviewer#1, Concern # 11: How to select and optimize the user-defined parameters in the proposed model? 

There is no general rule for it. Selecting the optimum number of the user-defined parameters is dependent on the performance of the model and the result of the evaluation metrics.

Reviewer#1, Concern # 12: There are quite a few abbreviations are used in the manuscript. It is suggested to use a table to host all the frequently used abbreviations with their descriptions to improve the readability

Thank you for your comment. We add a table of abbreviation to the manuscript. To quickly and easily check this table, it is copied below:

Abbreviation Definition

CNN Convolutional Neural Networks

CAD Computer-Aided Detection

ROI Region of Interest

EHMM Extended Hidden Markov Models

NS Neutrosophic Sets

F-FCM Fast Fuzzy C-Mean

SVM Support Vector Machine

KNN K-Nearest Neighbor

NN Naïve Bayes

PCA Principal Component Analysis

DNN Deep Neural Network

DWNN Deep-Wavelet Neural Networks

RELU Rectified Linear Activation Function

ADAM Adaptive Moment Estimation

SGDM Stochastic Gradient Descent with Momentum

RMSprop Root Mean Square propagation

TP True Positive

TN True Negative

FP False Positive

FN False Negative

GLCM Gray Level Co-occurrence Matrices

HOG Histogram of Oriented Gradients

n normal

ab abnormal

Reviewer#1, Concern # 13: Explain the evaluation metrics and justify why those evaluation metrics are used?

We edit evaluation the deep learning model section as following. To quickly and easily check this update in the Revised Manuscript with Track Changes page 11. Also, it is copied below:

Classification Metrics evaluate the performance of the model and measure how good or bad the classification is.

Accuracy: represents how many instances are completely classified correctly. It is calculated by dividing the total number of predictions by the number of right predictions. It is calculated by Eq. (3)

Accuracy=(T_P+T_N)/(T_P+T_N+F_P+F_N ) (3)

Sensitivity: is calculated based on how many patients have the disease are correctly estimated. It is calculated by Eq. (4)

Sensitivity=T_P/(T_P+F_N ) (4)

Specificity: is calculated based on how many patients do not have the disease are predicted right. It is calculated by Eq. (5)

Specificity=T_N/(T_N+F_P ) (5)

•True Positive (TP) refers to a positive-class sample that has been successfully classified by a model.

• False Positive (FP) refers to a sample that should have been classed as negative but was instead classified as positive.

• True Negative (TN) refers to a negative-class sample that has been successfully classified by a model.

• False Negative (FN) refers to a sample that should have been classed as positive but was instead classified as negative.

The accuracy metric indicates how many of the model's predictions were right. However, if the dataset is unbalanced, a model's high accuracy rate does not guarantee its ability to differentiate distinct classes equally. In specifically, in the classification of medical images, it is necessary to develop a model with the ability be applied to all classes. In cases, sensitivity and specificity should be used to provide information about the performance of the model. Sensitivity measures the percentage of patient have the disease that the proposed model correctly predicted. Specificity measures the percentage of patient do not have the disease and correctly estimated by the proposed model. These two evaluation metrics measure the ability of the model to decrease FN and FP predictions.

Reviewer#1, Concern # 14: Some sentences are too long to follow, it is suggested that to break them down into short but meaningful ones to make the manuscript readable

We try to break some sentences down into short as much as we can.

Reviewer#1, Concern # 15: The title is pretty deceptive and does not address the problem completely. 

The title of the article is changed into " Deep Learning Model for Fully Automated Breast Cancer Detection System from Thermograms"

Reviewer#1, Concern # 16: Every time a method/formula is used for something, it needs to be justified by either (a) prior work showing the superiority of this method, or (b) by your experiments showing its advantage over prior work methods - comparison is needed, or (c) formal proof of optimality. Please consider more prior works.

The limitations of the prior work are added to the literature review section. To quickly and easily check update in the Revised Manuscript with Track Changes page 7. Also, it is copied below:

From the discussed related work above, it could be remarked that the prior work has some limitations such as: 

(1) some related work used a small number of the dataset as in [6, 20]. 

(2) some related work did not consider segmentation of the breast area before classification such as in [27] or extract the breast area manually such as in [20, 21].

(3) some segmentation models such as in [1] removed parts of the breast. 

(4) some work has been evaluated by only calculating the accuracy metric only such as in [29]. However, the high accuracy rate of a model does not ensure its ability to distinguish different classes equally if the dataset is unbalanced [39]. 

Therefore, a fully automated breast cancer detection system from thermograms is needed and should be evaluated by not only the accuracy but also the most related metrics such as sensitivity and specificity. 

 Also, more recent prior works are added to the literature review section and compared with the proposed method in table 10. 

Reviewer#1, Concern # 17: The data is not described. Proper data description should contain the number of data items, number of parameters, distribution analysis of parameters, and of the target parameter itself for classification.

A data description table is added to the manuscript as table1. To quickly and easily check this table, it is copied below:

Dataset categories Dimension Training Validation Testing Total 

Normal 640x480 350 75 75 500

Abnormal 640x480 350 75 75 500

Reviewer#1, Concern # 18: The related works section is very short and no benefits from it. I suggest increasing the number of studies and add a new discussion there to show the advantage. Following studies can be considered

a. Breast-Cancer Detection using Thermal Images with Marine-Predators-Algorithm Selected Features.

b. Dilated Semantic Segmentation for Breast Ultrasonic Lesion Detection Using Parallel Feature Fusion.

c. Machine-Learning-Scheme to Detect Choroidal-Neovascularization in Retinal OCT Image

The recommended studies are cited in the manuscript as following:

a�23- Rajinikanth V, Kadry S, Taniar D, Damaševičius R and Rauf H T. Breast-Cancer Detection using Thermal Images with Marine-Predators-Algorithm Selected Features. In: 2021 Seventh International conference on Bio Signals, Images, and Instrumentation (ICBSII); Chennai, India. 2021.

b�30- Irfan R, Almazroi A, Rauf H, Damaševiˇcius R, Nasr E and Abdelgawad A. Dilated Semantic Segmentation for Breast UltrasonicLesion Detection Using Parallel Feature Fusion. Diagnostics.2021; 11: 1212.

c� 36- Rajinikanth V, Kadry S, Damaševičius R, Taniar D and Rauf H T. Machine-Learning-Scheme to Detect Choroidal-Neovascularization in Retinal OCT Image. In: 2021 Seventh International conference on Bio Signals, Images, and Instrumentation (ICBSII); Chennai, India. 2021.

Reviewer#1, Concern #19: Use Anova test to record the significant difference between performance of the proposed and existing methods.

We add statistical analysis section to the manuscript. To quickly and easily check this section in the Revised Manuscript with Track Changes page 16. Also, it is copied below:

4.7 Statistical Analysis

To analyze the evaluation of the proposed system statistically, we perform the analysis of variance (ANOVA) test, where the proposed system is compared with ResNet18, GoogleNet and VGG16 networks. The result of the ANOVA test is shown in table 9. To reject the null hypothesis, the p-value in the ANOVA test should be less than 0.05. According to Table 9, the p−value is less than 0.05. So, the null hypothesis was rejected by the results of ANOVA test.

 Results of the ANOVA test of the proposed model and CNN models

Model P-value

ResNet18 0.0423

GoogleNet 0.0173

VGG16 0.0023

Reviewer#2, Concern # 1: The image size is fixed as 228x228, but, the initial input dimension in UNet is depicted as 572x572. Please check and correct.

Thank you for comment. But, if you define the network, then you can change the input dimension in the input layer to your desired one. 

Reviewer#2, Concern # 2: The quality of the image is to be improved in every image cases.

Thank you for your comment. The quality of the image is improved

Reviewer#2, Concern # 3: What is the need for extracting the breast section, please justify. Further, include, how the CNN scheme is trained to segment the image (a clear discription is needed).

We add the answer of this question in Breast area segmentation using deep learning (CNN) section. To quickly and easily check this section in the Revised Manuscript with Track Changes page 8. Also, it is copied below:

The thermal image contains unnecessary areas as neck, shoulder, chess and other parts of the body which acts as noise during the training in CNN models. The aim of extracting the breast section is removing unwanted regions and using the areas destined to have cancer as the input of the CNN model for training and testing.

In U-Net, the initial series of convolutional layers are combined with max pooling layers to decrease the resolution of the input image. Then, these layers are followed by a series of convolutional layers combined with upsampling operators in order, so the resolution of the input image is increased. Combining these two paths produces a U-shaped graph that can used to perform image segmentation. For breast area segmentation with U-Net, Adaptive Moment Estimation (ADAM) method was used as optimized algorithm with number of epochs = 30. In addition, the training process was started with initial learning rate = 1.0e−3. The learning rate used a piecewise schedule and dropped by a factor of 0.3 every 10 epochs to allow the network to train quickly with a higher initial learning rate. The network trained with an 8-batch size to save memory.

Reviewer#2, Concern # 4: The image dimension, such as test image, Ground truth and segmented image is not looks like 228x228. Further in this database, the Ground Truth is not available. Please discuss, how the GT is generated (Hope, the test images are from:http://visual.ic.uff.br/dmi/).

Thank you for your comment. All images are resized to 228x228. Ground Truth images is not available for general in the site, we get it by personal communication. The test images are from the site.

Reviewer#2, Concern # 5 Please improve the reference section by considering few more related works.(Please refer, the following research works:

a. Breast-Cancer Detection using Thermal Images with Marine-Predators-Algorithm Selected Features

b. An Examination System to Classify the Breast Thermal Images into Early/Acute DCIS Class

c. A hybrid framework to evaluate breast abnormality using infrared thermal images

The recommended studies are cited in the manuscript as following:

a->23- Rajinikanth V, Kadry S, Taniar D, Damaševičius R and Rauf H T. Breast-Cancer Detection using Thermal Images with Marine-Predators-Algorithm Selected Features. In: 2021 Seventh International conference on Bio Signals, Images, and Instrumentation (ICBSII); Chennai, India. 2021.

b->20- Dey N, Rajinikanth V and Hassanien A. E. An Examination System to Classify the Breast Thermal Images into Early/Acute DCIS Class. In: Proceedings of International Conference on Data Science and Applications; Singapore. 2021.

c->10- Fernandes S, Kadry S and Rajinikanth V. A hybrid framework to evaluate breast abnormality using infrared thermal images. IEEE Consumer Electronics Magazine. 2019; 8: 31-36.

Reviewer#2, Concern # 6 I request you to compare the outcome of the proposed scheme with another results existing in the literature.

Thank you for your comment. This comparison is already existing in Table 10. Also, we compare the proposed scheme with the related work considered in comment 5 in the same table.

---

## [Decision Letter · Decision Letter 1]

23 Nov 2021

PONE-D-21-28729R1Deep Learning Model for Fully Automated Breast Cancer Detection System from ThermogramsPLOS ONE

Dear Dr. A. Mohamed,

Thank you for submitting your manuscript to PLOS ONE. After careful consideration, we feel that it has merit but does not fully meet PLOS ONE’s publication criteria as it currently stands. Therefore, we invite you to submit a revised version of the manuscript that addresses the points raised during the review process.

 Specifically, you need to address these concerns in the revised version:

- Improve the quality of all figures before submitting final version.

- Improve Fig. 10. The data is not discernable. Range Y-axis to [80,100]. Add the units of measurement (percentages).

- Include the data availability statement and the acknowledgement for ". " ext-link-type="uri" xlink:type="simple">http://visual.ic.uff.br/dmi/". 

We look forward to receiving your revised manuscript.

Kind regards,

Robertas Damaševičius

Academic Editor

PLOS ONE

Journal Requirements:

Additional Editor Comments:

Improve the quality of all figures before submitting final version.

Improve Fig. 10. The data is not discernable. Range Y-axis to [80,100]. Add the units of measurement (percentages).

Include the data availability statement and the acknowledgement for "" ext-link-type="uri" xlink:type="simple">http://visual.ic.uff.br/dmi/".

Reviewers' comments:

Reviewer's Responses to Questions

**Comments to the Author**

1. If the authors have adequately addressed your comments raised in a previous round of review and you feel that this manuscript is now acceptable for publication, you may indicate that here to bypass the “Comments to the Author” section, enter your conflict of interest statement in the “Confidential to Editor” section, and submit your "Accept" recommendation.

Reviewer #1: All comments have been addressed

Reviewer #2: All comments have been addressed

2. Is the manuscript technically sound, and do the data support the conclusions?

Reviewer #1: Yes

Reviewer #2: Yes

3. Has the statistical analysis been performed appropriately and rigorously? 

Reviewer #1: N/A

Reviewer #2: Yes

4. Have the authors made all data underlying the findings in their manuscript fully available?

Reviewer #1: Yes

Reviewer #2: Yes

5. Is the manuscript presented in an intelligible fashion and written in standard English?

Reviewer #1: Yes

Reviewer #2: Yes

6. Review Comments to the Author

Reviewer #1: The paper is well revised an can be accepted. However, please improve the quality of all figures before submitting final version.

Reviewer #2: Dear Authors,

The revised version of the paper is good.

If possible, include the data availability of include the acknowledgement for "" ext-link-type="uri" xlink:type="simple">http://visual.ic.uff.br/dmi/".

Thank you.

7. PLOS authors have the option to publish the peer review history of their article (what does this mean?). If published, this will include your full peer review and any attached files.

Reviewer #1: No

Reviewer #2: No

---

## [Author Response · Author response to Decision Letter 1]

21 Dec 2021

Original Manuscript ID: PONE-D-21-28729R1

Original Article Title: “Deep Learning Model for Fully Automated Breast Cancer Detection System from Thermograms “

To: PLOS ONE

Re: Response to reviewers

Dear Editor,

Thank you for allowing a resubmission of a revised version of the manuscript, with an opportunity to address the reviewers’ comments.

We are uploading (a) our point-by-point response to the comments (below) (response to reviewers), (b) an updated manuscript with yellow highlighting indicating changes, and (c) a clean updated manuscript without highlights 

Best regards,

Authors, 

Improve the quality of all figures before submitting final version.

Thank you for your comment. The quality of figures is improved 

Improve Fig. 10. The data is not discernable. Range Y-axis to [80,100]. Add the units of measurement (percentages).

Thank you for your comment. Figure 10 is improved as per your instruction.

Include the data availability statement and the acknowledgement for "http://visual.ic.uff.br/dmi/".

Thank you for your comment. The acknowledgement for (http://visual.ic.uff.br/dmi) is added to the paper. To quickly and easily check this, it is in page 20 of Revised Manuscript with Track Changes, it is also copied below:

Acknowledgement

The authors would like to thank the Department of Computer Science and the Hospital of the Federal University Fluminense, Niterói, Brazil, for providing DMR-IR benchmark database which is accessible through an online user-friendly interface (http://visual.ic.uff.br/dmi) and used for experiments.

---

## [Editor Report · Decision Letter 2]

22 Dec 2021

Deep Learning Model for Fully Automated Breast Cancer Detection System from Thermograms

PONE-D-21-28729R2

Dear Dr. Gaber,

We’re pleased to inform you that your manuscript has been judged scientifically suitable for publication and will be formally accepted for publication once it meets all outstanding technical requirements.

Kind regards,

Robertas Damaševičius

Academic Editor

PLOS ONE
---

## [Editor Report · Acceptance letter]

27 Dec 2021

PONE-D-21-28729R2 

Deep Learning Model for Fully Automated Breast Cancer Detection System from Thermograms 

Dear Dr. Gaber:

I'm pleased to inform you that your manuscript has been deemed suitable for publication in PLOS ONE. Congratulations! Your manuscript is now with our production department. 

Kind regards, 

on behalf of

Professor Robertas Damaševičius 

Academic Editor

PLOS ONE